# TremelImumab and Durvalumab Combination for the Non-OperatIve Management (NOM) of Microsatellite InstabiliTY (MSI)-High Resectable Gastric or Gastroesophageal Junction Cancer: The Multicentre, Single-Arm, Multi-Cohort, Phase II INFINITY Study

**DOI:** 10.3390/cancers13112839

**Published:** 2021-06-07

**Authors:** Alessandra Raimondi, Federica Palermo, Michele Prisciandaro, Massimo Aglietta, Lorenzo Antonuzzo, Giuseppe Aprile, Rossana Berardi, Giovanni G. Cardellino, Giovanni De Manzoni, Ferdinando De Vita, Massimo Di Maio, Lorenzo Fornaro, Giovanni L. Frassineti, Cristina Granetto, Francesco Iachetta, Sara Lonardi, Roberto Murialdo, Elena Ongaro, Francesca Pucci, Margherita Ratti, Nicola Silvestris, Valeria Smiroldo, Andrea Spallanzani, Antonia Strippoli, Stefano Tamberi, Emiliano Tamburini, Alberto Zaniboni, Maria Di Bartolomeo, Chiara Cremolini, Carlo Sposito, Vincenzo Mazzaferro, Filippo Pietrantonio

**Affiliations:** 1Department of Medical Oncology, Fondazione IRCCS Istituto Nazionale dei Tumori, 20133 Milano, Italy; alessandra.raimondi@istitutotumori.mi.it (A.R.); federica.palermo@istitutotumori.mi.it (F.P.); michele.prisciandaro@istitutotumori.mi.it (M.P.); maria.dibartolomeo@istitutotumori.mi.it (M.D.B.); 2Medical Oncology Unit, Candiolo Cancer Institute—FPO, IRCCS, 10060 Candiolo, Italy; massimo.aglietta@ircc.it; 3Clinical Oncology Unit, Careggi University Hospital, 50139 Florence, Italy; antonuzzol@aou-careggi.toscana.it; 4Department of Experimental and Clinical Medicine, University of Florence, 50121 Florence, Italy; 5Department of Clinical Oncology, Azienda ULSS 8 Berica-Ospedale di Vicenza, 36100 Vinceza, Italy; giuseppe.aprile@aulss8.veneto.it; 6Department of Clinical Oncology, Università Politecnica delle Marche—AOU Ospedali Riuniti di Ancona, 60126 Ancona, Italy; r.berardi@univpm.it; 7Department of Oncology, Presidio Ospedaliero “Santa Maria della Misericordia”—ASUFC, 33100 Udine, Italy; giovanni.cardellino@asuiud.sanita.fvg.it; 8Department of Surgery, Azienda Ospedaliero Universitaria Integrata di Verona—Borgo Trento, 37126 Verona, Italy; giovanni.demanzoni@univr.it; 9Medical Oncology Unit, Azienda Ospedaliero Universitaria dell’Università degli Studi della Campania “Luigi Vanvitelli”, 80131 Naples, Italy; ferdinando.devita@unicampania.it; 10Department of Oncology, Ospedale Mauriziano Umberto I, 10128 Turin, Italy; massimo.dimaio@unito.it; 11Medical Oncology 2, Azienda Ospedaliero Universitaria Pisana, 56126 Pisa, Italy; l.fornaro@ao-pisa.toscana.it; 12Department of Medical Oncology, Istituto Romagnolo per lo Studio dei Tumori “Dino Amadori”(IRST)-IRCCS, 47014 Meldola, Italy; luca.frassineti@irst.emr.it; 13Medical Oncology, Azienda Ospedaliera S. Croce e Carle, 12100 Cuneo, Italy; granetto.c@ospedale.cuneo.it; 14Department of Medical Oncology, Azienda Unità Sanitaria Locale—IRCCS di Reggio Emilia, 42122 Reggio Emilia, Italy; Francesco.Iachetta@ausl.re.it; 15Department of Clinical and Experimental Oncology, IRCCS Istituto Oncologico Veneto, 35128 Padua, Italy; sara.lonardi@iov.veneto.it; 16Medical Oncology, Ospedale Policlinico San Martino, 16132 Genoa, Italy; roberto.murialdo@hsanmartino.it; 17Department of Medical Oncology, Centro di Riferimento Oncologico di Aviano (CRO), IRCCS, 33081 Aviano, Italy; elena.ongaro@cro.it; 18Medical Oncology Unit, Azienda Ospedaliero Universitaria di Parma, 43126 Parma, Italy; fpucci@ao.pr.it; 19Oncology Department, Azienda Socio-Sanitaria Territoriale di Cremona, 26100 Cremona, Italy; mratti.cremona@gmail.com; 20Department of Molecular Medicine, Faculty of Advanced Medical Sciences, Tabriz University of Medical Sciences, 15731 Tabriz, Iran; n.silvestris@oncologico.bari.it; 21Medical Oncology Unit, IRCCS Istituto Tumori “Giovanni Paolo II” of Bari, 70124 Bari, Italy; 22Humanitas Cancer Center, Humanitas Clinical and Research Center, Rozzano, 20089 Milan, Italy; valeria.smiroldo@cancercenter.humanitas.it; 23Medical Oncology Unit, Azienda Ospedaliero Universitaria di Modena, 41125 Modena, Italy; spallanzani.andrea@aou.mo.it; 24Medical Oncology Unit, Policlinico Universitario A. Gemelli, 00168 Rome, Italy; antonia.strippoli@policlinicogemelli.it; 25Medical Oncology, Azienda USL Della Romagna, 48018 Faenza, Italy; stefano.tamberi@auslromagna.it; 26Medical Oncology Unit, Azienda Ospedaliera Cardinale G. Panico, 73039 Tricase, Italy; e.tamburini@piafondazionepanico.it; 27Department of Medical Oncology, Fondazione Poliambulanza, 25124 Brescia, Italy; alberto.zaniboni@poliambulanza.it; 28Department of Translational Research and New Technologies in Medicine and Surgery, University of Pisa, 56126 Pisa, Italy; chiaracremolini@gmail.com; 29Gastrointestinal Surgery and Liver Transplantation Unit, Fondazione IRCCS Istituto Nazionale dei Tumori, 20133 Milan, Italy; carlo.sposito@istitutotumori.mi.it (C.S.); vincenzo.mazzaferro@istitutotumori.mi.it (V.M.)

**Keywords:** gastric cancer, microsatellite instability, pre-operative treatment, PD-L1, CTLA4, non-operative management

## Abstract

**Simple Summary:**

The status of microsatellite instability (MSI-H) in gastric or gastroesophageal junction cancer (GC/GEJC) patients eligible for radical surgery proved to be prognostic for an improved survival outcome and predictive for poor/no benefit from the combination of adjuvant/peri-operative chemotherapy. MSI-H tumors display a high sensitivity to immunotherapy and exploratory studies showed that a pre-operative treatment with immune-checkpoint inhibitors may achieve elevated rates of pathological complete responses. The ongoing proof-of-concept INFINITY study is aimed at investigating the role of the combo-immunotherapy durvalumab plus tremelimumab as a neoadjuvant or potentially definitive treatment (avoiding surgery in case of complete clinical response) for MSI-H resectable GC/GEJC patients.

**Abstract:**

In resectable gastric or gastroesophageal junction cancer (GC/GEJC), the powerful positive prognostic effect and the potential predictive value for a lack of benefit from the combination of adjuvant/peri-operative chemotherapy for the MSI-high status was demonstrated. Given the high sensitivity of MSI-high tumors for immunotherapy, exploratory trials showed that combination immunotherapy induces a high rate of complete pathological response (pCR), potentially achieving cancer cure without surgery. INFINITY is an ongoing phase II, multicentre, single-arm, multi-cohort trial investigating the activity and safety of tremelimumab and durvalumab as neoadjuvant (Cohort 1) or potentially definitive (Cohort 2) treatment for MSI-high/dMMR/EBV-negative, resectable GC/GEJC. About 310 patients will be pre-screened, to enroll a total of 31 patients, 18 and 13 in Cohort 1 and 2, at 25 Italian Centres. The primary endpoint of Cohort 1 is rate of pCR (ypT0N0) and negative ctDNA after neoadjuvant immunotherapy, of Cohort 2 is 2-year complete response rate, defined as absence of macroscopic or microscopic residual disease (locally/regionally/distantly) at radiological examinations, tissue and liquid biopsy, during non-operative management without salvage gastrectomy. The ongoing INFINITY proof-of-concept study may provide evidence on immunotherapy and the potential omission of surgery in localized/locally advanced GC/GEJC patients selected for dMMR/MSI-high status eligible for radical resection.

## 1. Introduction

Gastric cancer (GC) and gastroesophageal junction cancer (GEJC) globally represent the fourth most common cancer and the second leading cause of cancer-related death [1]. Despite the evolution of the disease management thanks to the development of multimodality treatment strategies, GC/GEJC remains one of the most lethal malignancies with unsatisfactory long-term survival outcomes [2].

The cornerstone of potentially curative treatment remains surgery, specifically total/subtotal gastrectomy with lymphadenectomy. In the last decades, the research focused on the integration of systemic treatments for localized or locally advanced disease, in order to improve the survival outcome of patients and/or to increase the rate of radical surgical resections [3,4,5,6,7].

In detail, the MAGIC study first provided evidence that the combination of peri-operative chemotherapy with epirubicine, cisplatin and continuous fluorouracil (ECF) confers longer overall survival (OS) and a higher rate of curative resection than radical surgery alone [3]. Recently, the fluorouracil, oxaliplatin and docetaxel (FLOT) regimen showed improved survival outcomes as compared with ECF/ECX and is now considered the standard of care [4]. Otherwise, adjuvant chemotherapy in a GC setting was primarily supported by the ACTS-GC [6] and CLASSIC [5] studies, and the GASTRIC group meta-analysis [7]. Therefore, both chemotherapy approaches are now evidence-based and guideline-endorsed, the adjuvant schedule is preferred in Asian countries and the peri-operative one outside of Asia. Nevertheless, disease relapses still occur in a substantial number of patients, who eventually die for their disease and, on the other side, some patients are cured by surgery alone and do not require additional therapy, thus they receive potentially toxic treatments without a significant benefit [8,9].

The mismatch repair deficient (dMMR)/microsatellite instability high (MSI-H) status is widely recognized as a prognostic and predictive factor in colorectal cancer (CRC) [10,11]. Recently, in the setting of resectable GC/GEJC, the powerful positive prognostic effect for improved disease-free survival (DFS) and OS and the potential negative predictive value for a lack of benefit from the combination of adjuvant/peri-operative chemotherapy in MSI-H patients was shown [12]. Moreover, the importance of dMMR/MSI-H status emerged when it proved to represent one of the most powerful predictors of benefit from immunotherapy with anti-programmed death receptor 1 and its ligand (PD1/PD-L1) immune-checkpoint inhibitors (ICIs) so that pembrolizumab received the agnostic approval for adult and pediatric patients with advanced tumors bearing this alteration [13,14].

This evidence provided the rationale for the investigation of neoadjuvant immunotherapy in MSI-H tumors, aimed at maximizing the rate of disease eradication and the chance of cure for patients. The exploratory and proof-of-concept trials conducted so far, mainly in CRC patients, showed that the combination of anti-PD1/PD-L1 with anti-cytotoxic T-lymphocyte antigen 4 (CTLA4) agents is able to induce a very high rate of complete pathological responses (pCR) in patients selected for dMMR/MSI-H status, potentially achieving a curative power aside from surgery [15,16,17,18].

On this basis, the INFINITY study was designed, aimed at assessing the activity and safety of the combination of the anti-CTLA4 tremelimumab and the anti-PD-L1 durvalumab as a neoadjuvant or potentially definitive treatment for MSI-H, dMMR and EBV negative, early-stage and resectable GC/GEJC.

## 2. Materials and Methods

### 2.1. Study Design

INFINITY is a phase II, multicentre, single-arm, multi-cohort trial conducted in Italy. Patients affected by early-stage and resectable GC/GEJC selected for the presence of MSI-H, dMMR and EBV negative status centrally determined at the Co-ordinating Centre are eligible for the study.

Following the initial complete baseline disease staging, patients are administered a neoadjuvant treatment with combination immunotherapy composed of tremelimumab and durvalumab and, after the complete tumor restaging, they undergo the standard surgery according to the clinical practice and guidelines. Afterward, patients are followed in a standard follow up phase. After the completion of the enrollment in Cohort 1 and the extensive evaluation of the final results of Cohort 1 regarding all endpoints (including exploratory endpoints), and after potential amendment(s) on study design, eligibility criteria and study procedures requested by the Sponsor’s Steering Committee and an Independent Data Monitoring Committee made of foreign experts, and following the approval of the Ethics Committee and the Italian Medicines Agency, the enrollment in Cohort 2 will start. Patients enrolled in Cohort 2 will receive the same neoadjuvant treatment as in Cohort 1 and will be subjected to complete disease restaging; those with no evidence of complete clinical response will be subjected to standard surgery and subsequent follow up as in Cohort 1. Patients with complete clinical response will undergo a non-operative management (NOM), followed by standard follow-up in absence of disease recurrence/persistence, with salvage surgery in case of evidence of disease. The study design is depicted in Figure 1 and Figure 2.

### 2.2. Study Population

The study will be conducted in 25 Italian Cancer Centres and approximately 310 patients will be molecularly pre-screened, in order to enroll a total number of 31 patients, 18 in Cohort 1 and 13 in Cohort 2.

The main inclusion criteria are:written informed consent and any locally required authorizationage ≥18 years oldEastern Cooperative Oncology Group (ECOG) Performance Status (PS) 0–1body weight >30 kgdiagnosis of resectable GC or GEJC (Siewert II-III), categorized according to TNM classification 8th edition as cT ≥2, any cN, M0 or any cT, cN1-3, M0absence of distant metastases as defined by negativity of computed tomography (CT) and 18-fluorodeoxyglucose positron-emission tomography (18-FDG PET)MSI-H status confirmed by immunohistochemistry (IHC) and multiplex polymerase chain reaction (PCR) and EBV-negative status confirmed by means of silver in-situ hybridization (SISH for EBER), as determined centrally at the Co-ordinating Centre with lack of heterogeneity of dMMR status as showed by lack of tumor cells showing concomitant expression of all 4 protein markersadequate bone marrow and organ function.The main exclusion criteria account for:signs of distant metastasesprior medical treatments or irradiation for GC/GEJCmajor surgical procedure within 28 days prior to the first dose of treatmentprevious treatments with anti-CTLA4 or anti-PD1/PD-L1 ICIsallergy or severe hypersensitivity reaction to monoclonal antibodiesautoimmune diseases or history of organ transplantation that require immunosuppressive therapyactive primary immunodeficiencyactive infection including tuberculosis, hepatitis B, hepatitis Cuncontrolled intercurrent illness, or psychiatric illness/social situationswomen in pregnancy or lactation condition.

### 2.3. Study Endpoints

The primary objective of the study is to assess the activity of the combination immunotherapy with tremelimumab plus durvalumab as a neoadjuvant or definitive treatment of resectable MSI-H GC/GEJC. Therefore, the primary endpoint of Cohort 1 is the rate of pCR (ypT0N0) and negative ctDNA status after neoadjuvant immunotherapy in the intention-to-treat population. The primary endpoint of Cohort 2 is the 2-year complete response rate, defined as the absence of macroscopic or microscopic residual disease (locally, regionally and distantly) at radiological examinations, tissue and liquid biopsy, in absence of salvage gastrectomy.

Secondary endpoints are, for both Cohorts, 3-year DFS, defined as time from the enrollment in the study to the occurrence of disease relapse (local and/or distant), second GC/GEJC primary, or death from any cause; 5-year OS, defined as time from the enrollment in the study to the occurrence of death; metastases-free survival, defined as time from the enrollment in the study to the first evidence of metastases or death from any cause; the safety of the neoadjuvant combination immunotherapy, defined as the incidence of AEs during the treatment and follow-up phases, assessed according to common terminology criteria for adverse events (CTCAE v5.0), and its impact on patients’ quality of life, assessed thanks to patient reported outcomes (EORTC QLQ-C30, EORTC QLQ-STO22 and EuroQol EQ-5D, during pre-operative treatment phase). Additionally, for Cohort 2 the gastrectomy-free survival, defined as time from the inclusion in the study to the occurrence of gastrectomy or death from any cause, and for patients enrolled in Cohort 1 and for those enrolled in Cohort 2 and subjected to surgery, morbidity and mortality of gastrectomy following pre-operative tremelimumab and durvalumab.

Exploratory endpoints will involve translational analyses on tissue and blood biomarkers aimed at identifying which subgroup of patients may derive the highest chance of definitive cure from immunotherapy combination of tremelimumab plus durvalumab.

### 2.4. Statistical Design

Regarding Cohort 1, considering a pCR (ypT0N0) rate of 50% or less as non-acceptable for a subsequent NOM strategy (H0 = 50%) and hypothesizing a target pCR rate of 80% following neoadjuvant tremelimumab and durvalumab, 18 patients will be required to be able to reject the null hypothesis of unacceptable pCR rate, with one-sided α- and β-errors of 0.05 and 0.20, respectively. The study treatment will be deemed promising if at least 13 pCRs will be observed.

Regarding Cohort 2, considering as non-acceptable a 2-year clinical-pathological-molecular complete response of 50% after definitive treatment and hypothesizing a target rate of 85% following tremelimumab and durvalumab, 13 patients will be required to be able to reject the null hypothesis of unacceptable rates of long-term cure, with one-sided α- and β-errors of 0.05 and 0.20, respectively. The study treatment will be deemed promising if at least 10 clinical-pathological-molecular complete responses will be observed at 2 years in absence of salvage gastrectomy.

### 2.5. Study Procedures

Patients meeting the eligibility criteria undergo the pre-screening phase, in which the tumor specimens are centrally evaluated at the Co-ordinating Centre and only patients with MSI-H and dMMR status confirmed according to both IHC and multiplex PCR and EBV-negative status confirmed by SISH technique may be enrolled in the study.

In both cohorts, patients are subjected to a baseline complete staging as per standard practice with chest-abdomen-pelvis CT with contrast and 18-FDG PET, endoscopic ultrasonography (EUS) as per national and international guidelines, diagnostic laparoscopy, if clinically indicated as per judgment of the Investigators and as per national and international guidelines [8,9], with the addition of liquid biopsy for detecting specific mutations, amplifications or translocations in circulating tumor DNA (ctDNA) and plasma/PMBCs collection for exploratory analyses. Subsequently, patients receive the ICIs combination for a 12-week period, tremelimumab 300 mg in single administration (day 1) and durvalumab 1500 mg every 4 weeks for 3 cycles (day 1, day 29, day 57). Between weeks 12 and 14, patients undergo a complete disease restaging, with chest-abdomen-pelvis CT with contrast, 18-FDG PET/CT, EUS with multiple random biopsies of the tumor site and fine needle aspiration biopsy (FNA) of clinically suspicious regional nodes and liquid biopsy.

In Cohort 1, all patients undergo standard gastrectomy with D2 lymphadenectomy between weeks 15 and 18 from enrollment (at least six weeks after the last treatment administration). Afterward, patients undergo a phase of follow-up with visits every 12 weeks according to the standard clinical practice and local guidelines, during the first two years or until death or disease recurrence. After the first two years, in absence of disease local or distant recurrence, patients terminate the phase of protocol-scheduled follow-up and are monitored every 6 months until the fifth year from surgery according to the standard clinical practice and local guidelines. In Cohort 2, patients who do not achieve clinical response undergo standard surgery, as in Cohort 1, while in presence of complete response confirmed by imaging/EUS, pathology and liquid biopsy, patients are subjected to a NOM and start a protocol-scheduled follow-up phase with follow-up visits repeated every 12 weeks for two consecutive years. During each follow-up visit patients undergo chest-abdomen-pelvis CT with contrast, 18-FDG PET/TC, if clinically indicated by the judgment of the investigator and not mandatory, EUS with multiple random biopsies of the tumor site and FNA of clinically suspicious regional nodes, liquid biopsy for the assessment of the presence of minimal residual disease in ctDNA. A study monitoring committee including three independent multidisciplinary teams established at the participating institutions will review the clinical records of each follow-up visit by means of an interactive video-conference system and will make decisions on the management of individual patients. At any time during the follow-up, in case of clinical suspicion or confirmation of residual GC, either at the imaging, pathologically at tissue biopsies/cytological specimens or at ctDNA in liquid biopsy, patients undergo standard surgery according to the clinical practice at their Centre. After the first two years of follow-up, in absence of disease local or distant recurrence, patients terminate the active follow-up phase and are monitored every 6 months until the fifth year according to the standard clinical practice and local guidelines. Further details are provided in Table 1, Table 2 and Table 3.

## 3. Discussion

The treatment decision-making for resectable GC/GEJC is currently based upon the clinical and pathological staging, in absence of validated biomarkers potentially able to select patients eligible for the combination with adjuvant/peri-operative chemotherapy or surgery alone [19].

The dMMR/MSI-H status is a well-established prognostic factor for prolonged survival in stage II-III CRC patients, and a potential predictive marker for the lack of benefit from adjuvant fluoropyrimidine monotherapy in stage II disease. Therefore, the indication to test for MMR/MSI status in all patients operated for stage II-III colon cancer and to consider this biomarker during the clinical management is widely accepted and recommended by all the major guidelines [10,11]. Moving to the setting of curatively resected GC/GEJC, the results of two post-hoc analyses of pivotal randomized clinical trials, MAGIC and CLASSIC studies, suggested a similar relationship between MSI-H status and patients outcomes, even though with limited statistical power due to the low prevalence of MSI-H in GC (8–10% of resectable GC patients enrolled in clinical trials) [20,21]. Afterward, an Individual Patient Data pooled analysis combining the results of four large international randomized trials in resectable GC (MAGIC, CLASSIC, ARTIST and ITACA-S) was performed, in order to evaluate the interaction between MSI status and the outcome of patients in a well-powered meta-analysis. The results confirmed the powerful positive prognostic effect of MSI-H status in surgically resected GC patients and the lack of benefit of peri-operative or adjuvant chemotherapy after surgery in this molecularly-selected subgroup [12].

Nowadays, immunotherapy revolutionized the therapeutic algorithm and provided unprecedented improvements in patients’ outcomes in several tumors, so that it now represents the standard of care in a large number of cancer types such as melanoma, non-small cell lung cancer (NSCLC) and kidney cancer. Considering GC and GEJC, the clinical trials conducted in an unselected population provided globally unsatisfactory results, highlighting the need for a molecular selection of patients according to predictive biomarkers of response to immunotherapy [22,23,24,25]. In fact, the phase III, randomized, open-label KEYNOTE-061 study, investigating pembrolizumab versus paclitaxel as second-line treatment in advanced GC or GEJC progressing after platinum and fluoropyrimidine, did not meet its primary endpoint of improving OS in the overall population, even though pembrolizumab showed a better safety profile [24]. Consistently, the KEYNOTE-062 trial compared a first-line treatment with pembrolizumab alone or combined with chemotherapy to the standard chemotherapy in advanced HER2 negative GC or GEJC, selected for PD-L1 Combined Positive Score (CPS) ≥1 and ≥10. In the CPS ≥1 population, pembrolizumab monotherapy resulted to be non-inferior to chemotherapy in terms of OS, while pembrolizumab plus chemotherapy did not demonstrate superior OS and progression-free survival (PFS) as compared to standard chemotherapy [25].

The importance of dMMR/MSI-H status emerged since these tumors, across different primary sites of origin, proved to be highly responsive to immunotherapy, to the extent that Food and Drug Administration granted an accelerated approval to the anti PD-1 antibody pembrolizumab for adult and pediatric patients with agnostic unresectable/metastatic dMMR or MSI-H cancers [13]. This was based on the analysis of the outcome of 149 selected patients enrolled in 5 clinical trials (KEYNOTE-016, 164, 012, 028, 158), among which 5 were affected by GC or GEJC. The key finding was that the objective response rate to pembrolizumab was 62% and 60% in dMMR CRC and non-CRC cancers, respectively, as compared to 0% in MMR proficient (pMMR) CRCs [13,26]. Afterward, the clinical trials introducing ICIs in the first-line treatment for dMMR/MSI-H mCRCs showed unprecedented results, setting immunotherapy as a new standard of care in this selected population. In details, in the Checkmate 142 trial, the combination of nivolumab (3 mg/kg every 2 weeks) plus low-dose ipilimumab (1 mg/kg every 6 weeks), both given until disease progression or unacceptable toxicity, reached a 64% and 58% overall response rate (ORR) and 84% and 78% disease control rate (DCR) per investigator assessment and blinded central review, respectively. Moreover, median PFS and OS were not reached at the 19.9 months median follow up, with impressive 15-month PFS and OS rates of 75% and 84% [27]. Consistently, the phase III, randomized, open-label KEYNOTE-177 trial demonstrated the superiority of pembrolizumab as compared to standard chemotherapy in terms of PFS (median PFS 16.5 vs. 8.2 months, *p* = 0.0002) with a reduced toxicity burden [28]. Considering GC or GEJC, the exploratory analyses of the KEYNOTE-059 and -061 and -062 trials showed that patients with MSI-H GC had a dramatic benefit in terms of response and survival outcomes from immunotherapy. In particular, in Cohort 1 of KEYNOTE-059, where the ORR with pembrolizumab monotherapy was 11.6% in the overall trial population, an objective response was observed in 57.1% MSI-H patients versus 9.0% of MSS ones [23]. In the KEYNOTE-061 study, patients with MSI-H tumors, irrespectively of PD-L1 CPS, had a median OS not reached with pembrolizumab versus 8.1 months with paclitaxel, and 7/15 patients (47%) achieved an objective response with pembrolizumab versus 2/12 (17%) with paclitaxel [24]. Finally, in the KEYNOTE-062 trial, in the MSI-H subgroup median OS was not reached versus 8.5 months, median PFS was 11.2 versus 6.6 months, and ORR was 57.1% versus 36.8% with pembrolizumab versus standard chemotherapy, respectively [25]. These data, together with the global burden of evidence collected so far, highlight how immunotherapy-based combinations could represent a novel standard of care for patients with MSI-H metastatic GC [14].

Moving forward, although MSI-H tumors account for about 10% of locally advanced GCs, and they display a poor or absent benefit from chemotherapy while being endowed with intrinsic sensitivity to immunotherapy, the conduction of dedicated trials with immunotherapy in the setting of MSI-H GC may be of paramount clinical relevance for a high number of cancer patients. For what concerns the use of ICIs as a neoadjuvant treatment, recent studies showed that pre-operative immunotherapy could achieve a high rate of pathologic major or complete response in potentially resectable neoplasms and eventually provide a chance to cure the tumor regardless of surgery. In fact, the priming of robust anti-tumor T cell responses necessitates the presence of an intact tumor mass [29], which provides abundant neoantigens essential for the interactions between tumor cells and immune cells [30,31]. This necessity is evidenced by the inability of ICIs to establish protective immunity after radical operations [32]. Furthermore, neoadjuvant immunotherapy results in an early establishment of immunologic memory which is absent in the adjuvant setting, contributing to the elimination of micro-metastases that otherwise may cause relapse [33]. First of all, in NSCLC, a well-known immunotherapy-responsive tumor, Forde et al. showed that two doses of the anti-PD-1 nivolumab (3 mg/kg every two weeks) in untreated, resectable (stage I-IIIA) tumors was safe and did not delay the planned surgery, while conferring a major pathological response in 9 out of 20 patients operated irrespectively of PD-L1 status, although the pre-treatment tumor mutational burden was predictive of pathologic response [34]. Afterward, Chalabi et al. [15] in the phase II NICHE study tested the combination of ipilimumab (1 mg/kg on day 1) and nivolumab (3 mg/kg on days 1 and 15) for 1 cycle as a “window of opportunity” treatment in two parallel cohorts of dMMR or pMMR resectable colon cancers. While no meaningful pathological response was reported in the 20 patients with MSS cancers, significant activity was shown in terms of disease eradication in the 20 patients with MSI-H tumors, of whom 81% had clinical stage III disease, where pCR was confirmed in 12/19 (63%) cases while all but one of the remaining cases showed a major pathological response [15]. More recently, Zhang et al. [18] reported a case series of six patients with resectable, locally advanced (cT4N+) and MSI-H gastric (*n* = 4) or colon (*n* = 2) cancers treated with anti-PD1-based regimens, even immediately after a rapid progression to standard chemotherapy. After radical surgery, 5 out of 6 patients (87%) achieved a pCR, whereas the single patient with pathological downstaging in absence of pCR had a heterogeneous mixed dMMR-pMMR cancer (a scenario previously reported in 8% of cases by Kim et al. [17]). Finally, Ludford et al. [16] published the results of a retrospective series of patients with stage IV metastatic MSI-H CRC, showing pCR in 13 out of 14 resected metastases and even after a short-duration therapy. This study highlights that immunotherapy, particularly anti-PD1 plus anti-CTLA4 combination, may provide cure even for patients with metastatic disease, consistently with the plateau of PFS curves in clinical trials with dual checkpoint inhibition (2-year PFS of about 70%) [35].

On the basis of the evidence collected, as furtherly illustrated in Table 4, and on the strong biological and clinical rationale above-discussed, the INFINITY study was designed, to evaluate the activity and safety of combination immunotherapy as a neoadjuvant or potentially definitive treatment for resectable GC/GEJC selected for MSI-H and dMMR.

The study treatment regimen consists of an anti-CTLA4 agent (tremelimumab) and an anti-PD-L1 agent (durvalumab) since the mechanisms of action of CTLA4 and PD-1/PD-L1 checkpoints are non-redundant, thus targeting both PD-1/PD-L1 and CTLA4 pathways may have an additive or even synergistic activity [36]. In fact, the combination of durvalumab plus tremelimumab proved to be safe and effective in different tumor settings such as NSCLC, head and neck tumors and recently CRC, since it demonstrated to prolong survival with manageable tolerability and preserved quality of life in metastatic CRC patients unselected for MMR status [37]. The treatment schedule of a single priming dose of tremelimumab (300 mg flat dose) combined with three doses of durvalumab (1500 mg flat dose q 4 weeks) was selected since the recently-presented D4190C00022 study showed that this schedule is endowed with the best benefit-to-risk profile and increases the proliferative CD8-positive T cells, potentially associated with tumor response [38]. This result was in line with the proven dose-dependent benefit for CTLA4 inhibitors in combination with anti-PD1/PD-L1 and with the evidence that an initial dose-dependent burst of CD4-positive T cells can be mobilized to peripheral blood after the first anti-CTLA4 agent infusion, not recurring with repeated dosing [39,40].

The single-arm multi-cohort trial design is justified considering that, given the excellent 5-year DFS (70%) and OS (75%) of patients with radically resected MSI-H GC, clinical trials comparing a neoadjuvant/peri-operative immunotherapy approach with standard of care multimodality strategy (surgery +/− neoadjuvant chemotherapy) would require a very high number of patients to be enrolled to demonstrate a significant increase of survival above the outcomes historically achieved by surgical resection. The feasibility of such a trial is even impaired by the relatively low prevalence of MSI-H status in GC (about 9% in the recent meta-analysis [12]). Finally, even when hypothesizing positive and practice-changing results of such a trial, the majority of patients eligible for peri-operative immunotherapy would be overtreated since they are most likely cured with surgery alone. Therefore, given the significant morbidity and mortality of gastrectomy, and its potentially negative impact on long-term outcomes in terms of quality of life, there is a strong rationale for investigating PD-(L)1 plus CTLA4 blockade as a definitive treatment and NOM strategy in MSI-H GC in Cohort 2 based on the results of Cohort 1 of the INFINITY Study.

## 4. Conclusions

In conclusion, the ongoing INFINITY proof-of-concept study may provide biologically sound clinical evidence on the therapeutic management of localized/locally advanced GC/GEJC patients selected for dMMR/MSI-H status eligible for radical resection, specifically for what concerns the role of neoadjuvant immunotherapy with ICIs and the potential omission of radical surgery.

## Figures and Tables

**Figure 1 cancers-13-02839-f001:**
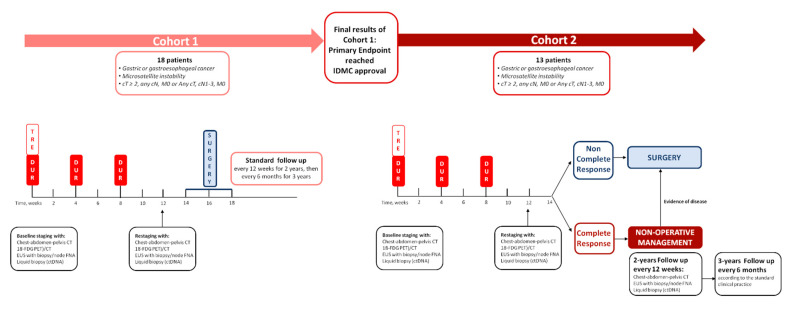
This figure illustrates the INFINITY study design.

**Figure 2 cancers-13-02839-f002:**
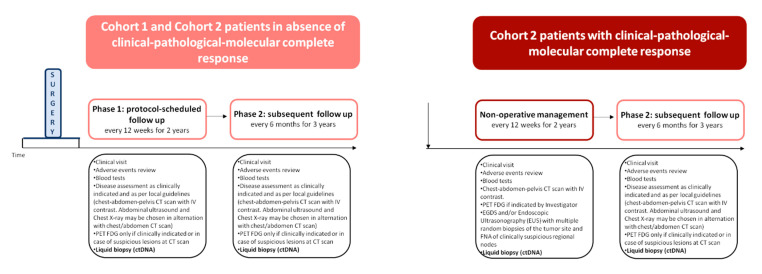
This figure illustrates the follow up phase of the INFINITY study.

**Table 1 cancers-13-02839-t001:** Schedule of assessments for all pre-operative immunotherapy arms (Cohort 1 and 2).

Procedures	Pre-Screening	Screening	Treatment Period			Restaging and Response Assessment
**Treatment cycle**	**Week -6 to -4**	**Week -4 to -1**	**C1**	**C2**	**C3**	**Weeks 12–14**
**Day**		**-28 to -1**	**1**	**29**	**57**	**85–99**
**Window (days)**	**NA**	**NA**	**(±3 days assessments)**	**(±3 days, tumor assessments and PRO assessments ±7 days)**		
**Informed consent**						
Informed consent to pre-screening	X					
Informed consent main: study procedures		X				
**Study procedures**						
Full physical exam (including height)		X				
Targeted physical exam (based on symptoms)			X	X	X	X
Vital signs (including weight)		X	X	X	X	X
12-leads ECG		X	as clinically indicated			
Concomitant medications		<-------------------------------- All visits --------------------------->				
Baseline characteristics		X				
Eligibility criteria	X	X	X			
**Laboratory assessments**						
Serum chemistry		X	X	X	X	X
Hematology		X	X	X	X	X
Coagulation		X				X
Biomarkers: CEA, CA 19.9		X				X
Thyroid function (TSH, free T3, free T4)		X		X	X	X
**Virology assessment**						
HBsAg, anti-HBs, anti-HBc, anti-HCV and HBV DNA		X				
HIV		X				
Urinalysis		X				X
Pregnancy test		X	X	X	X	X
**Tumor assessments**						
Disease assessment by chest-abdomen-pelvis CT scan with intravenous contrast		X				X
18-FDG PET/CT scan		X				X (if clinically indicated)
Endoscopic ultrasonography (EUS) with multiple random biopsies of the tumor site and FNA of clinically suspicious regional nodes		X				X
Microsatellite instability (MSI) centralized assessment	X					
Diagnostic laparoscopy		X				
**Other assessments and assays**						
Liquid biopsy		X				X
Exploratory biomarkers blood sample and PBMCs		X	X	X	X	X
**Monitoring**						
ECOG performance status		X	X	X	X	X
AE/SAE assessment		<----------------------------- All visits ----------------------------->				
**Study drug administration**						
Tremelimumab 300 mg			X			
Durvalumab 1500 mg			X	X	X	
**PRO assessments**						
PROs		X		X	X	X

X signs are placed in boxes corresponding to the time scheduled for each assessment and procedure.

**Table 2 cancers-13-02839-t002:** Schedule of assessments for Cohort 1 and for Cohort 2 patients in absence of clinical complete response.

Evaluation	Surgery	Follow-Up	
**Time from enrolment**	**Week 15–18**		
		**Phase 1: Protocol-scheduled follow-up Every 12 weeks for 2 years**	**Phase 2: Subsequent follow-up** **Every 6 months for 3 years**
**Study procedures**			
Full physical exam	X	X	
Targeted physical exam (based on symptoms)			X
Vital signs (including weight and BMI)	X	X	X
12-leads ECG	X	as clinically indicated	
Concomitant medications	X	<---------------------- All visits ---------------------->	
Management of post-surgical complications ^c^		X	X
**Laboratory assessments**			
Serum chemistry	X	X	X
Hematology	X	X	X
Biomarkers CEA, CA 19.9	X	X	X
Thyroid function (TSH, free T3, free T4)	X	X	as clinically indicated
Urinalysis	X	as clinically indicated	as clinically indicated
Pregnancy test	X	Until the third month since last dose of study drugs	
**Tumor and disease assessments**			
Disease assessment by means of chest-abdomen-pelvis CT scan with intravenous contrast. Abdominal ultrasound and Chest X-ray may be chosen in alternation with chest/abdomen CT scan		As clinically indicated and as per local guidelines	As clinically indicated and as per local guidelines
18-FDG PET/CT scan		Only if clinically indicated or suspicious lesions at CT scan	Only if clinically indicated or suspicious lesions at CT
EGDS +/- Endoscopic Ultrasonography (EUS) with fine needle aspiration (FNA) biopsy of suspicious lesions		As clinically indicated and as per local guidelines	
**Other assessment and assays**			
Liquid biopsy (ctDNA in plasma)	X	X	X
Exploratory biomarkers blood sample and PBMCs	X	X	X
**Monitoring**			
ECOG performance status	X	X	X
AE/SAE assessment	X		
AE assessment		as clinically indicated	

**Table 3 cancers-13-02839-t003:** Schedule of assessments for Cohort 2 patients with clinical complete response.

Evaluation	Non-Operative Management (NOM) Strategy	
**Time from enrolment**	**Week 15–18**	
	**Phase 1: Protocol-scheduled follow-up Every 12 weeks for 2 years**	**Phase 2: Standard follow-up** **Every 6 months for 3 years**
**Study procedures**		
Full physical exam	X	
Targeted physical exam (based on symptoms)		X
Vital signs (including weight and BMI)	X	X
ECG	as clinically indicated	
Concomitant medications	<------------------------ All visits ------------------------>	
**Laboratory assessments**		
Serum chemistry	**X**	X
Hematology	X	X
Biomarkers CEA, CA 19.9	X	X
Thyroid function (TSH, free T3, free T4)	X	as clinically indicated
Urinalysis	X	as clinically indicated
Pregnancy test	Until the third month since last dose of study drugs	
**Tumor and disease assessments**		
Disease assessment by means of chest-abdomen-pelvis CT scan with intravenous contrast	X	
Disease assessment by means of chest-abdomen-pelvis CT scan with intravenous contrast. Abdominal ultrasound and Chest X-ray may be chosen in alternation with chest/abdomen CT scan		as clinically indicated and as per local guidelines
18-FDG PET/CT scan	if clinically indicated by the judgment of the Investigator	Only if clinically indicated or suspicious lesions at CT scan
EGDS +/- Endoscopic Ultrasonography (EUS) with multiple random biopsies of the tumor site and FNA of clinically suspicious regional nodes	X	If clinically indicated
**Other assessment and assays**		
Liquid biopsy	X	X
Exploratory biomarkers blood sample and PBMCs	X	X
**Monitoring**		
ECOG performance status	X	X
AE assessment	as clinically indicated	

**Table 4 cancers-13-02839-t004:** Literature-based evidence on immunotherapy in MSI-H tumors.

Study	Disease Setting	Treatment	Results
KEYNOTE 059 [23]	Advanced previously treated GC	Pembrolizumab monotherapy	Overall response rate 11.6% overall trial population 57.1% MSI-H patients 9.0% MSS patients
KEYNOTE 061 [24]	Advanced GC progressing after platinum and fluoropyrimidine	Pembrolizumab vs. paclitaxel	*Overall population* Median OS 9.1 vs. 8.3 months (HR 0.82, 95% CI 0.66-1.03; *p* = 0.0421) Median PFS 1.5 vs. 4.1 months (HR 1.27, 95% CI 1.03-1.57)
			*MSI-H subgroup* Median OS not reached vs. 8.1 months (HR 0.42; 95% CI 0.13–1.31)
KEYNOTE 062 [25]	Advanced previously untreated GC	Pembrolizumab vs. chemotherapy	*Overall population* Median OS 10.6 vs. 11.1 months in CPS ≥1 (HR 0.91; 99.2% CI 0.69–1.18) while 17.4 vs. 10.8 months in CPS ≥10 (HR 0.69; 95% CI 0.49–0.97)
			*MSI-H subgroup* Median OS not reached vs. 8.5 months Median PFS 11.2 vs. 6.6 months
		Pembrolizumab plus chemotherapy vs. chemotherapy	Median OS 12.5 vs. 11.1 months in CPS ≥1 (HR 0.85; 95% CI 0.70–1.03; *p* = 0.05) while 12.3 vs. 10.8 months in CPS ≥10 (HR 0.85; 95% CI 0.62–1.17; *p* = 0.16)
Analysis of KEYNOTE-016, 164, 012, 028, 158 trials [13,26]	Mismatch repair deficient (dMMR) colorectal cancers (CRCs), dMMR non-CRC cancers, pMMR CRCs	Pembrolizumab	Objective response rate 62% in dMMR CRC 60% in dMMR non-CRC cancers 0% in pMMR CRCs
CheckMate 142 [27]	Metastatic previously untreated MSI-H CRC	Nivolumab plus low dose ipilimumab	Overall response rate 64% (58% per BICR) Disease control rate 84% (78% per BICR) Median PFS not reached 15-month PFS rate 75% Median OS not reached 15-month OS rate 84%
KEYNOTE 177 [28]	Metastatic previously untreated MSI-H CRC	Pembrolizumab vs. standard chemotherapy	Median PFS 16.5 vs. 8.2 months (HR 0.60; 95% CI 0.45–0.80; *p* = 0.0002) Estimated restricted mean survival after 24 months of follow-up 13.7 (range 12.0–15.4) vs. 10.8 months (range 9.4–12.2)

## Data Availability

The study is ongoing. Data regarding the study will be available on motivated request from the corresponding author since data would not be publicly available due to privacy restrictions.

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
