# Peer review of "TremelImumab and Durvalumab Combination for the Non-OperatIve Management (NOM) of Microsatellite InstabiliTY (MSI)-High Resectable Gastric or Gastroesophageal Junction Cancer: The Multicentre, Single-Arm, Multi-Cohort, Phase II INFINITY Study"

_cancers, 2021, doi:10.3390/cancers13112839_

Round 1

Reviewer 1 Report

The authors present their manuscript describing the design of the ongoing single arm INFINITY trial with Cohort 1 being composed of neoadjuvant treatment with tremelimumab + durvalumab prior to surgery in MSI-H/dMMR/EBV negative gastric cancer.  Provided their predefined rate of successful pathologic complete responses are observed, their trial will shift to a second Cohort 2 exploring non-operative management following the same tremelimumab + durvalumab treatment in this good prognosis population when treated surgically.  The manuscript is overall well written with possible further benefit from minor English editing, but the authors cite the requisite literature for the background and rationale supporting the design of their trial.  Central confirmation of the requisite MSI-H/dMMR/EBV negative status is an advantage of the study design to minimize the potential of inaccurate biomarker testing.  Successful achievement of their primary endpoints may further improve survival in this patient population and possibly quality-of-life if patients can achieve biologic cures in the absence of surgery.

Author Response

We thank the Reviewer for his/her comments upon our manuscript, we agree with the potential impact of the trial results upon the clinical management of patients affected by localized or locally advanced gastric/gastroesophageal junction cancer selected for MSI-H/dMMR/EBV negative status, for what concerns the improvement of the survival outcome and of quality of life.

We performed a linguistic editing of the paper as requested.

Reviewer 2 Report

This is a protocol by Raimondi et al. describing a multicentre, single-arm, multi-cohort study assessing the activity of tremelimumab plus durvalumab as neoadjuvant or definitive treatment of resectable MSI-H gastric cancer or gastroesophageal junction cancer. The study consists of two cohorts, with all participants enrolling in Cohort 1 after complete baseline staging and receiving neoadjuvant neoadjuvant combination immunotherapy with tremelimumab and durvalumab is given before standard surgery. After complete restaging, they will be enrolled in Cohort 2 receiving the same neoadjuvant treatment as Cohort 1, with those with complete clinical response receiving non-operative management and those with no evidence of complete clinical response receiving standard surgery.

The protocol is comprehensive with each stage and procedures involved in the study well-described. Literature is adequate, featuring clinical evidence that supports the basis of the study.

Study design and population

This cohort study aims to enroll a total of 31 patients, with 18 in Cohort 1 and 13 in Cohort 2. According to the statistical design, at least 13 pCRs must be observed in Cohort 1 and 13 patients in Cohort 2 in order to reject the null hypothesis.

The inclusion and exclusion criteria is comprehensive based upon the patient factors and clinical parameters listed. It may be preferable to illustrate these inclusion and exclusion criteria in a table or point-form given the current length in prose.

Study endpoints

Currently, Figure 1 which illustrates the study design include the details of Cohorts 1 and 2 and the investigations part of baseline staging and restaging.

Given the detail of follow-up and observation as part of the secondary endpoints in this study, it would be good to illustrate this timeline in a similar figure.

Study procedures

Study procedures are comprehensively described and well-substantiated. It would be useful to have the table of procedures included.

Discussion

The discussion is well supported with literature and past clinical evidence. The evidence in suggestion of using the combination immunotherapy that is part of this study

This protocol cites a series of past trials studying MSI-H malignancies, both gastrointestinal and extra-gastrointestinal. Given the volume of evidence, it would be preferable to summarise the most pertinent findings in prose while additional details could be left in a table.

Author Response

This is a protocol by Raimondi et al. describing a multicentre, single-arm, multi-cohort study assessing the activity of tremelimumab plus durvalumab as neoadjuvant or definitive treatment of resectable MSI-H gastric cancer or gastroesophageal junction cancer. The study consists of two cohorts, with all participants enrolling in Cohort 1 after complete baseline staging and receiving neoadjuvant neoadjuvant combination immunotherapy with tremelimumab and durvalumab is given before standard surgery. After complete restaging, they will be enrolled in Cohort 2 receiving the same neoadjuvant treatment as Cohort 1, with those with complete clinical response receiving non-operative management and those with no evidence of complete clinical response receiving standard surgery.

The protocol is comprehensive with each stage and procedures involved in the study well-described. Literature is adequate, featuring clinical evidence that supports the basis of the study.

We thank the Reviewer for his/her comments upon our manuscript.

Study design and population

This cohort study aims to enroll a total of 31 patients, with 18 in Cohort 1 and 13 in Cohort 2. According to the statistical design, at least 13 pCRs must be observed in Cohort 1 and 13 patients in Cohort 2 in order to reject the null hypothesis.

The inclusion and exclusion criteria is comprehensive based upon the patient factors and clinical parameters listed. It may be preferable to illustrate these inclusion and exclusion criteria in a table or point-form given the current length in prose.

As requested, we implemented the manuscript with the description of inclusion and exclusion criteria in a point-form.

Study endpoints

Currently, Figure 1 which illustrates the study design include the details of Cohorts 1 and 2 and the investigations part of baseline staging and restaging.

Given the detail of follow-up and observation as part of the secondary endpoints in this study, it would be good to illustrate this timeline in a similar figure.

We added the manuscript Figure 2 that reports the timeline of the follow-up and observation phase.

Study Procedures

Study procedures are comprehensively described and well-substantiated. It would be useful to have the table of procedures included.

As specifically requested, we provided the tables of the study procedures in Table 1, Table 2 and Table 3.

Discussion

The discussion is well supported with literature and past clinical evidence. The evidence in suggestion of using the combination immunotherapy that is part of this study.

This protocol cites a series of past trials studying MSI-H malignancies, both gastrointestinal and extra-gastrointestinal. Given the volume of evidence, it would be preferable to summarise the most pertinent findings in prose while additional details could be left in a table.

We added Table 4 as requested.
